# Palestine Energy Policy for Photovoltaic Generation: Current Status and What Should Be Next?

**Tamer Khatib \***[ID]**, Amin Bazyan, Hiba Assi and Sura Malhis**

Department of Energy Engineering and Environment, An-Najah National University, Nablus 97300, Palestine; a.bazyan@stu.najah.edu (A.B.); h.assi@stu.najah.edu (H.A.); s.malhis@stu.najah.edu (S.M.)
\* Correspondence: t.khatib@najah.edu

**Abstract:** Most of the consumed energy in Palestine comes from Israel. Meanwhile, the Israeli government controls the amount of electricity for Palestinians due to political reasons. This has led to many electricity shortages, prompting the Palestinians to invest in grid connected photovoltaic systems to mitigate electricity shortages. However, the lack of experience and loose energy policies have negatively affected the electricity distribution network in Palestine. Thus, this paper aims to discuss the current energy policy model for photovoltaic generation in Palestine and the challenges facing it. Moreover, 15 photovoltaic systems are selected in this research for technical and economical evaluation, to first show the typical performance of photovoltaic systems in Palestine, and second, to prove that there are failure cases in many systems due to a number of behavioral and structural barriers. Finally, the paper proposes a suggestion of unbundling transmission lines in the region to address the current critical status of photovoltaic investment in Palestine. As a result, the typical average yield factor of photovoltaic systems in Palestine is in the range of 1368–1816 kWh/kWp per year with a payback period of 5.5–7.4 years. However, the percentage of failure for the installed systems is found to be 47%. Meanwhile, the low awareness and lack of non-technical information are the main behavioral barriers, while grid infrastructure, lack of technical standards and staff training as well as loose and discouraging policies are the most dominant structural barriers.

**Keywords:** photovoltaic; yield factor; capacity factor; feasibility; energy policy; Palestine

## 1. Introduction

Palestinian territories clearly suffer from the scarcity of the conventional energy sources, high population growth and rising prices of energy [1]. Thus, this would lead Palestine to a developing energy crisis. In 2018, Palestine's total energy demand reached around 5800 GWh, in which the Israel Electric Company (IEC) covered around 92.6% of this demand. The rest of the energy supplies are from Jordan (1.5%), Egypt (0.6%), Gaza Power Plant (4.4%) [2]. Meanwhile, renewable energy sources accounting for 10.2% respectively of the total demand in 2018 [2]. The high energy imports from the IEC had left the Palestinian Authority (PA) with an estimated debt of 574 million USD [3]. On the other hand, the cost of energy (CoE) is relatively high in Palestine, where CoE is approximately 0.6215 ILS/kWh for residential sector. Meanwhile, the CoE in Israel is approximately 0.4516 ILS/kWh [3]. It is expected that seasonal power shortages will be emerging in the West Bank following a demand growth of 3.5% per year until 2030.

Palestine has some potential of renewable energy sources that could make a change for the whole situation. For instance, Palestine has an estimated annual average daily solar energy in the range of (5.4 kWh/m$^2$–6 kWh/m$^2$) with sunshine hours over 3000 h per year. However, this average daily solar energy goes as low as 2.6 kWh/m$^2$ in December and becomes up to 8.4 kWh/m$^2$ in June [3–9]. Based on that, the PA, through Palestinian Energy and Natural Resources Authority (PENRA) has set a number of policies for encouraging investment in photovoltaic (PV) systems. Moreover, the Palestine Investment Promotion Fund (PIF) which is a public body connected to the PA, had set PV systems as Palestine's

major investment opportunities for local and international investors, with an estimated market size of 50 million USD [10]. Additionally, international programs, such as SUNREF had been launched in the past few years to encourage utilizing PV systems in Palestine through soft loans. Finally, many countries, worldwide, have started to install PV systems in Palestine as a form of support to the development process. For example, the Czech Republic is accounted for installing about 0.5 MWp of PV systems through scattered projects [3–9]. As a fact, the funding of PV system projects by international funding agencies has become a recent goal. Following that, many companies and institutions started planning for PV projects by applying funding proposals for international funding agencies [11–13]. In [12], a list of many projects is provided with an interesting analysis on the performance of these systems. The authors have claimed the payback period of such projects is in the range of 5 to 8.5 years [12]. However, the authors of [12] only studied systems with normal operation as the scope of the paper is purely technical.

Energy policy is a dominant factor that may cause the success or failure of photovoltaic generation. In [1], a review of a good energy policy that promotes PV generation in many countries is proposed. The review of these good examples of energy policies aims to provide a general frame for PV generation polices. According to [14], good photovoltaic generation must take into consideration good matching between future plans and market capacity, reasonable tax reduction, dissemination of technical and non-technical information, enforcing financials and fiscal incentives as well as setting standards for the technology so as to guarantee best possible performance. These policy features are seconded by [15] as well, where good photovoltaic generation policies were analyzed for three countries which are Germany, Japan and China. Similarly, in [16], the authors proposed the analysis of energy policies for photovoltaic generation in the EU. The authors claimed that there are some positive polices in this regard, such as the feed in tariff. Meanwhile, the regulatory uncertainty is considered as the major negative energy policy.

On the other hand, in [17], the authors measured end-users perceptions and satisfaction of photovoltaic technology India so as to show the low acceptance level of this technology. After that the authors analyzed the enforced polices by the government so as to show the current barriers for such a technology and suggest solutions for policy amendments so as to promote this technology.

In general, the feature of any energy policy depends on the system in the country, political situations, culture, level of income, geographical location and many other factors. Thus, specific analysis for each country is required. However, this analysis can be based on good policies selected in other countries subject that there is a similarity in countries characteristics. An example of this important analysis is provided in [6], where the energy policy for Columbia is analyzed based on general policy features and consideration of the characteristics of the country itself. In this research a review of installed projects, plans and energy situation is presented first, then analysis of the selected policies is done. Finally, conclusions and recommendations were provided based on the previous analysis so as to better promote PV generation.

In fact, many researchers have proposed theoretical based studies which boost the interest in investing in grid-connected photovoltaic systems by government, municipality buildings and private firms [11]. This interest is justifiable by many theoretical studies such as [6], where the authors have conducted an analysis for rooftop PV systems and assumed that if 10% of each Palestinian rooftop has an average area of 150 m$^2$, this will lead a total energy yield of 146 GWh per year and that would cover approximately 2.5% of all electricity imports in Palestine. A similar theoretical study is presented in [18], where the author proposed a large PV system to be connected to the grid with very promising numbers such as yield factor and payback period without considering real grid infrastructure and the impact of such a proposal on the grid.

Following that, many projects have been implemented in Palestinian cities. In summary, there are a 39 MW installed PV systems, while there are 93 MWp of PV systems under development. Moreover, there are about 24 MWp PV systems proposed officially

for approval, with several MWp of PV system in planning (detailed list is provided later). However, most of these grid-connected systems have an energy yield of less than expected. Moreover, despite this huge amount of installed PV systems, a significant reduction of energy demand from Israel was not noticed in Palestine. Based on that, this paper aims to discuss the current energy policy model for photovoltaic generation in Palestine. The model is reviewed and analyzed so as to show the current challenges that it faces. Moreover, 15 photovoltaic systems were selected for technical and economical evaluation in this research so as to show first the typical performance of photovoltaic systems in Palestine. Second, the analysis is done to prove the failure of many systems due to many behavioral and structural barriers. These barriers are also discussed and analyzed in this research. Finally, a suggestion of unbundling transmission lines in the region is proposed so as to address the current critical status of renewable energy investment in Palestine. As a result, the typical average yield factor of photovoltaic systems in Palestine is 1368–1816 kWh/kWp with a payback period in the range of 5.5–7.4 years. Meanwhile, the low awareness and lack of non-technical information are the main behavioral barriers, while, grid infrastructure, lack of technical standards and staff training as well as loose and discouraging policies by the governments are the most structural barriers.

## 2. Energy Frame Work, Stakeholder and Policy Model in Palestine

According to Oslo accords, any development in the field of electricity or energy should be in collaboration between Palestine and Israel. However, the current political situation is banning any collaboration in this regard. Meanwhile, as mentioned before, most of the consumed energy in the west bank comes from Israeli power generation stations that are located nearby the borders of the West Bank and Gaza. In general, the power which is generated at the Israeli side is transmitted via 161 kV main transmission lines. These lines then get transformed to 33 kV transmission lines or 22 kV transmission lines. Then in the middle of these transmission lines, a coupling point that controls the amount of electricity that flows toward the Palestinian side is placed. These coupling points are controlled by Israeli side while the Palestinian side does not have any role in operating these coupling points. There are about 200 coupling points in the West Bank of such a type. These coupling points have caps on the electricity current where the ampacity is controlled by an entity called Israeli civil administration of the West Bank. Any increase in the ampacity is done based on political coordination.

The Israeli Electricity Company (IEC) owns these transmission lines up to the coupling point while the Palestinian Electricity Transmission Line Company possess the remaining part until the low voltage power substation which can be 33 kV/0.4 kV, 22 kV/0.4 kV or 33 kV/6.6 kV. After the low voltage substation, the electricity networks are either possessed by private Palestinian electricity distribution companies (DESCOs) or local councils. As for Gaza, there is a power station that generates power to some regions of Gaza, while there is a 161 kV connection with Israel as well as a 260 kV connection with Egypt. However, currently most of the electricity is being delivered by the 161 kV transmission line following the damage of the Gaza power plant in 2014 during the war on Gaza.

Recently, the Palestinian Authority (PA) has been trying to develop their energy framework so as to be able to manage this sector in a better way. To do so, the PA has established the Palestine Electricity Transmission Line Company (PETL) so as to hold the responsibility of some 161 and 33 kV transmission lines and to regulate the relation with the IEC through it. Figure 1 shows the desired Palestinian institutional energy framework. Based on this model, DESCOs are responsible of supplying electricity to the end users. Each DESCO is responsible of a geographical area. However, some areas are still served by private distribution networks operators who deal and buy electricity directly from IEC. These private networks are owned by either municipalities or local councils and mainly located in villages. Figure 2 below shows the demands and location of all DESCOs in Palestine.

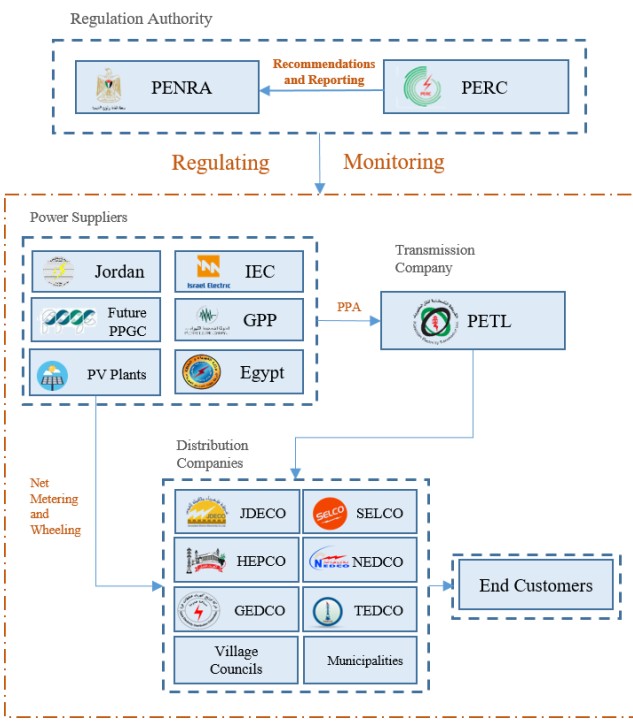

**Figure 1.** Desired Palestinian institutional energy framework.

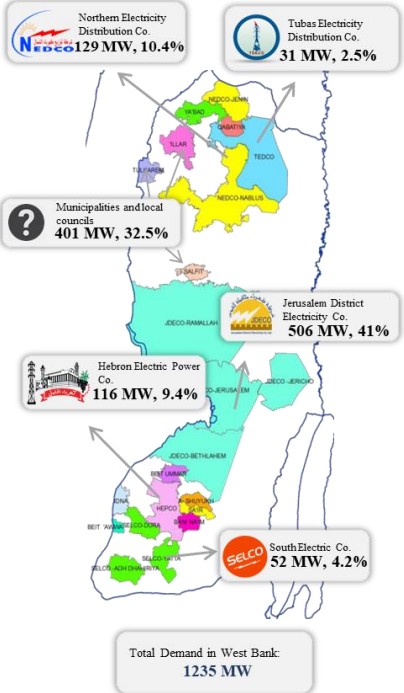

**Figure 2.** Electricity demand per private Palestinian electricity distribution company (DESCO) in the West Bank.

However, the model illustrated in Figure 1 is far from the real situation due to many challenges for PETL. In fact, PETL cannot operate very well yet due to political reasons. In general, the West Bank contains many Israeli settlements. These settlements are powered by the same 161 kV lines that power the Palestinian regions. Thus, PETL cannot take responsibility of these lines as it will be powering Israeli settlements which are illegal according to UN resolutions. In the meanwhile, Israel will never allow a public Palestinian

company to control the power flow to Israeli settlements. Moreover, PA is still unable to control and unify Palestinian distribution network operators due to the nature of these entities. Some of these entities are private, while some of them are public but not owned by the government such NEDCO, and TEDCO, while some of them are Jordanian such as JEDCO. Other operators are municipalities and local councils which are directly related to the ministry of local government where PENRA has no authority over them.

PA actually has been established in 1994, while most of power distribution operators where in business much before that, in some cases since 1914 like JDECO. Meanwhile, due the political situation these operators where not nationalized (kept private or semi-private) to protect them from the Israeli occupation of the West Bank and Gaza. Here when the PA come in 1994, it did not show any intention to appropriate or nationalize any of the Palestinian entities as it was an authority, and such an action should be done when the state is officially declared and recognized so as to protect the state's capabilities.

Thus, when it comes to the real energy policy model, the framework illustrated in Figure 1 becomes more complicated with multidirectional political relations and control. For example, in Figure 1, the IEC is assumed to be regulated and monitored by the Palestinian institution, which is actually not the case. Therefore, the real energy policy model with the actual intitule energy framework is illustrated in Figure 3. Meanwhile Table 1 lists the energy stakeholders in Palestine.

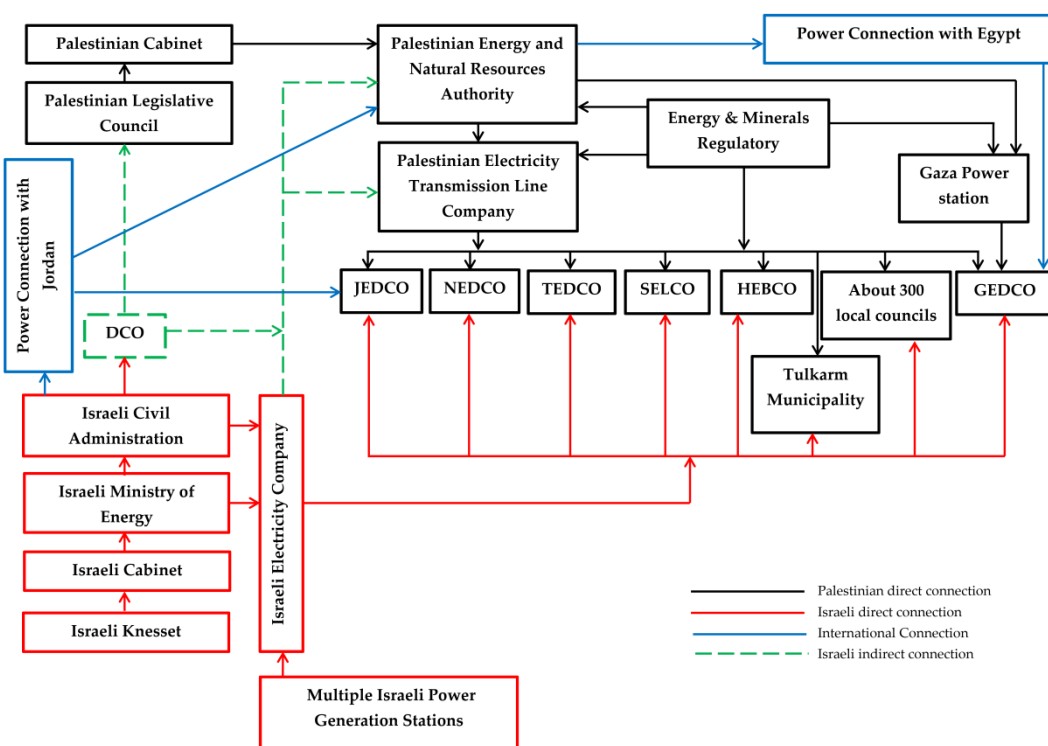

**Figure 3.** Current situation of the energy framework in Palestine.

**Table 1.** List of energy stakeholders in Palestine.

| Entity | Description, Rules and Connections |
|---|---|
| Palestinian Legislative Council (PLC) | It is an elected body to represent the people of Palestine and it is in charge of approving any Act regarding electricity. This entity has a direct relation with Palestinian Cabinet. However, the council is being suspended nowadays due to political reasons, while the Presidential office is taking charge of its responsibility via presidential acts |
| Palestinian Cabinet (PC) | It implies the head of PENRA, and has two rules, first to propose act drafts to the PLC, and to direct PERNA to implement the approved acts via agreed rules and laws. PC may have an indirect relation with DCO and Israeli civil administration |
| PENRA | It is the head of electricity policy model and it acts as ministry of energy in fact. PENRA is being regulated by Energy and Minerals Regulatory and it has direct connections with all model's components. PENRA is responsible of managing this model, paying electricity bills to Israel, managing international issues, conducting negotiations in this regard and managing the local issues with all electricity distributers |
| Energy and Minerals Regulatory | This Regulatory should monitor the rules and regulations and help in drafting it including shares, rules, market, concession areas and prices |
| Jerusalem District Electricity company (JDECO) | Responsible of distributing electricity for Ramallah, Jericho, Bethlehem, East Jerusalem and some other villages. They have direct connection with IEC, DCO, PENRA, PETL as well as Jordan |
| North Electricity Distribution company (NEDCO) | Responsible of distributing electricity for Nablus Jenin, Salfet and some other villages. They have direct connection with IEC, DCO, PENRA as well as PETL |
| Tubas Electricity Distribution company (NEDCO) | Responsible of distributing electricity for Tubas and some other villages. They do have direct connection with IEC, DCO, PENRA as well as PETL |
| Southern Electricity company (SELCO) | Responsible of distributing electricity for southern region of Hebron and some other villages. They do have direct connection with IEC, DCO, PENRA as well as PETL |
| Hebron Electricity company (HEBCO) | Responsible of distributing electricity for Hebron city and some other villages. They do have direct connection with IEC, DCO, PENRA as well as PETL |
| Gaza Electricity Distribution company (GEDCO) | Responsible of distributing electricity for Gaza. They do have direct connection with IEC, PENRA, Egypt, as well as PETL |
| Tulkarm Municipality | Responsible of distributing electricity for Tulkarm city. They do have direct connection with IEC, DCO, PENRA as well as PETL |
| Local councils | They have responsibility to distribute electricity to villages as individual entity and they do have direct connection with IEC, DCO, PENRA as well as PETL |
| Gaza power station | Responsible of generating power to some parts of Gaza. They do have direct connection with Israeli civil Administration, DCO, PENRA, Egypt, Qatar as well as PETL |
| DCO | It is a coordination office to manage issues including electricity between PA and Israel |
| PETL | Electricity transmission company that aims to possess transmission line to regulate the relation between distribution companies and IEC through it |

　　　　In addition to that, there are many other countries and international organizations which donate and fund photovoltaic power system to Palestine as a support of the development process in the country. These counties are also considered as stakeholders and actually can be included in the illustrated energy policy model in Figure 3. These countries and international organizations are—IFC: International Finance Corporation, The World

Bank, USAID, PIF: Palestinian Investment Fund, European Bank for Reconstruction and Development, EIB: European Investment Bank, GIZ: German Society for International Cooperation, AFD: The French Development Agency, Netherlands Rep. Office, The Czech Republic, Italy, CCC: Consolidated Contractors Corporation, JICA: Japan International Cooperation Agency, The Republic of China, ISDB: Islamic Development Bank.

### 3. Laws and Regulation for PV Systems in Palestine

The PA has managed to regulate the energy sector since it was established in 1994. This can be seen by different acts and laws enforced by the council of ministers to improve the sector. Firstly, the Council released Decision No. 13 in 2009 by the name "The General Electricity Law". This law was established to initiate the regulation and upgrading of the electricity sector in Palestine. This law had been modified several times to handle any related problems through Decision No. 16 in 2012 and Decision No.17 in 2018. Next, Decision No. 9 in 2010 which was confirmed by Decision No. 6/14/14 in 2012 that was published to set the general rules and laws for electric companies to operate and function. As for renewable energy, the Council announced Decision No. 14 in 2015, which gave a thorough definition of what is renewable energy, encouraged utilizing renewable energy and determined the tasks of all parties related to this sector, such as the transmission company, distribution companies and research centers. Additionally, Decision No.11/79/17 handled utility-scale renewable energy plants in terms of tariff and purchase agreements. Finally, Decision No. 6 in 2017 indicated the PA's major interest in solar energy as this decision specifically handles the incentives for installed PV systems as can be seen in Table 2. For utility-scale PV plants that exceed 1 MWp, taxes reduction was introduced to encourage the private sector to invest in such projects, since such incentives increase the profitability of such systems.

**Table 2.** Income taxes incentives for utility-scale PV projects.

| Stage | Income Tax Value for Solar-Produced-Energy Sales | Period |
|:---:|:---:|:---:|
| 1 | 0% | 7 years from the date of running the station |
| 2 | 5% | 5 years from the end of stage 1 |
| 3 | 10% | 3 years from the end of stage 2 |
| 4 | Normal income taxes values are considered (15%) | End of stage 3 |

Meanwhile, there are special incentives for net-metering projects owned by Palestinian Investment Fund (PIF). PIF is a public fund owned by PA. PV systems projects are one of the activities of PIF, and thus, special incentives are given to projects owned and managed by PIF. These incentives are similar for the incentives illustrated in Table 2 but with longer periods as listed below:

- Projects with a capacity of 20 kWp are given additional year for all stages;
- Projects with a capacity of 40 kWp are being given two more years for all stages (almost the whole lifetime);
- Projects with a capacity of 60 kWp are being given three more years for all stages (almost the whole lifetime);
- Any other PIF project that do benefit from the previous incentives, and its capacity have been developed to be up to 40 kWp at least, can benefit from an income tax of (2%) for two years.

On the other hand, purchase prices of electrical energy produced from solar energy projects are applied according to Table 3 based on its capacity and classification.

**Table 3.** Solar energy tariff per scheme.

| Station's Capacity and Classification | Purchase Prices (USD/kWh) |
|---|---|
| Domestic (within the Palestinian Initiative for solar energy, up to 5 kWp and for the first 1000 house). | (0.14–0.15) (same as normal electricity price for end-user) |
| Net-metering (bigger than 5 kWp and up to 999 kWp) | There is no specified purchasing tariff, those projects are treated as (unit versus unit), and according to the regulations of each project. PENRA defines "net metering" as a policy that forces DESCos to buy from PV system owners any excess electricity they may generate. In this case the energy meter runs backwards to provide credits for a whole year. Afterwards, the credit is set to zero at the beginning of April of each year. The consumer is then billed only based on the net electricity usage for each month. |
| Generation plants with a capacity of 1MW–5MW (offered by the investor to the government) | 90% of the production will be purchased based on agreements with PETL. The price will be the average purchase price of various sources of conventional energy. |
| Generation plants with a capacity of 1–5 MW (called by the government and offered according to bidding process) | The lowest price among the competitive investors without exceeding 90% of the average purchase price from various sources of conventional energy. |

Prices in the previous table are regularly revised, when the technology prices face changes, when the capacity of the projects exceeds 56 MW or when the purchase prices of conventional energy changes. Moreover, all the facilities that sell PV system related equipment and tools are exempted from customs duties.

## 4. Evaluation Criteria for the Selected PV System

In order to evaluate the performance and the feasibility of PV system investments in Palestine, technical and economic factors are taken into account. The technical evaluation implemented in this paper consists of three main indicators which are yield factor (YF), capacity factor (CF) and performance ratio (PR). YF is used to compare the regular energy production of the project to its total peak power. This factor can be calculated by the following equation:

$$YF = \frac{E_{pv} \ (kWh \ Production)}{P_{PV \ peak} \ (Rated \ kWp)} \tag{1}$$

In general, the evaluation of YF can be done based on reference values for the country or nearby countries. In general, such a value should be generated based on the performance of actual system installed in the country and it should be generated for every year to understand the impact of the aging factor on the YF. Meanwhile, many factors affect the value of the yield factor as it mainly depends on solar radiation that reaches the collector, which is directly connected to the collector's tilt angle and orientation. Secondly, the module's sensitivity to high temperatures and low light levels can also affect the YF. This is to say that different brands of PV panel can have different YF reference values. Finally, the losses occurring in the system and its efficiency also affect the YF.

On the other hand, CF indicates the relation between the output energy of a system for a specific period, to the energy of that system at its maximum power potential (actual power generated divided by the theoretical maximum power). CF can be calculated as follows:

$$CF = \frac{Epv \ (kWh/monthly)}{12*P(kWp)*30} \tag{2}$$

CF is a well know ratio in power system engineering and in general it indicates the usage of the power source. Thus, a capacity factor of 100% means that the source is fully utilized. For PV system there is also a reference capacity factor. However, CF value is a general reference value not like YF.

Finally, the performance factor is a very important measure of PV projects since it evaluates the quality of the PV system without considering the incident solar irradiation or the orientation of the solar panels.

It is agreed to define the performance factor as 'the ratio of the actual energy output of the PV system to the expected energy output'. The following formula is used for manual calculation of the performance factor:

$$\text{P.R} = \frac{\text{Actual reading of plant output in kWh}}{\text{Calculated, nominal plant output in kWh}} \tag{3}$$

In general, there are many factors that would affect the value of the performance factor. The first factor is the cell temperature of the PV module, as high cell temperature values affect the efficiency of the PV panel negatively and consequently the production of the system. In addition to that, power dissipation, conduction losses through the system, inverter's efficiency and the quality of system's installation and equipment's affect the PF. Thus, this factor also shows the quality of installed systems.

On the other hand, the economical assessment includes both costs and benefits of the system. To reach this result, economical parameters like the simple pay-back period (SPP) is selected. Simple payback period is simply defined as the time taken for the revenues from the PV project to equal the capital cost that is spent in the project's investment. SPP can be calculated by:

$$\text{SPP} = \frac{\text{Investment}}{\text{Revenues}} \tag{4}$$

To propose a fair economical assessment, real expenditure which refers to the gross costs the selected PV projects are needed. Thus, real costs are taken into consideration. These costs are obtained from the contractors who installed the selected PV systems in 2016. In general, the cost of the kWp of PV system in 2016 was found to be 1750 USD.

## 5. Results of PV System Evaluation

In this research 15 PV systems were selected. These PV systems are ranging from 5 kWp–35 kWp. These systems are either owned by private companies or installed based on donation from foreign funding agencies. The installation period of these systems was 2015–2016. Meanwhile the period of the evaluation was from January 2018 to December 2018 (365 days). All of the evaluated systems are grid connected systems with a scheme of net meeting. As mentioned before, the cost of the kWp is 1750 USD, while operation and maintenance costs are assumed to be zero as it is not defined or taken into account when the feasibility study was done. It is worth to mention that 70% of these systems are funded by EU funding agency to governmental and local sociality bodies while 30% of these systems are owned by private companies (motivations will be discussed later). It is also important to mention that the methodologies followed in calculating these metrics are depending on the assumption and methodologies followed by PV system operators here in Palestine, and thus, it is not necessary to reflect the real performance of PV system perfectly. However, to discuss the failure of these systems, the current assumptions that are used by PV operators are utilized.

Figure 4 shows the monthly yield factor for these systems. The results are varying in the figure, and this might be because of a technical problem or a consequence of a misbehavior (a discussion follows later on this point). Anyway, the highest annual yield factor is found to be 1816 kWh/kWp while, the lowest is found to be 685 kWh/kWp. This shows that the assumption followed in Palestine that the 1 kWp generates 1750 kWh per year is very loose and cannot be utilized to design optimum PV system.

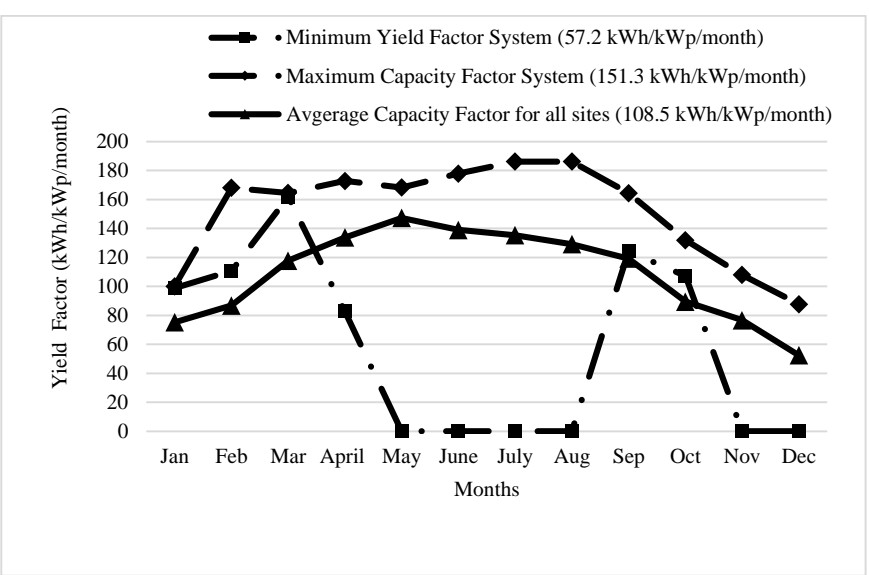

**Figure 4.** Yield factor (YF) for the selected photovoltaic (PV) system.

On the other hand, Figure 5 shows the monthly capacity factor of the selected systems, where the maximum CF of a PV power is found to be 50%. After all, the value of CF of selected system is found in the range of (2–50%) (Monthly basis). Here, these results also show real problems with these systems, that are due to technical reasons that will be discussed later in this paper.

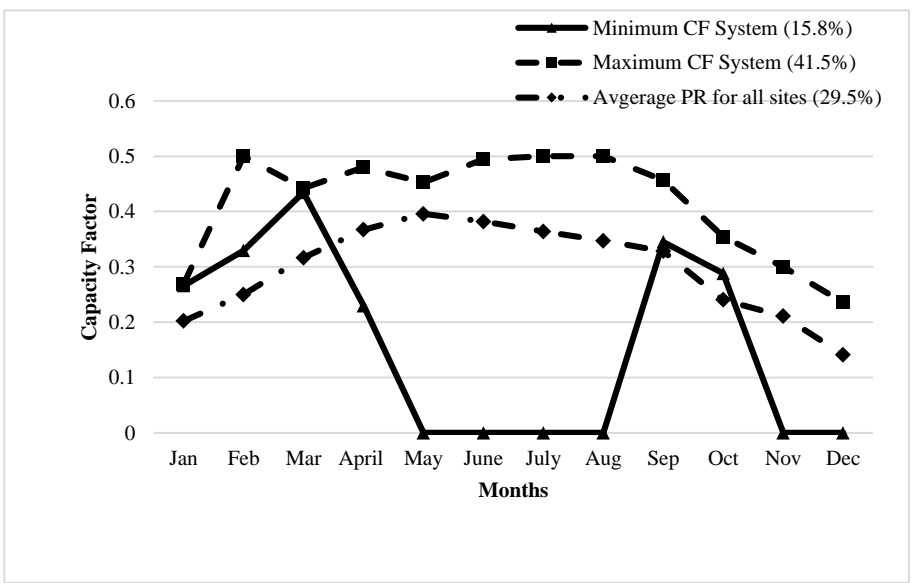

**Figure 5.** Monthly capacity factor for the selected PV systems.

Finally, PR averages are shown in Figure 6. From Figure 6 it is very clear that the PR values have a lot of problems where they are fluctuating all over the year. Meanwhile, for a PV system the PR is expected to be almost the same for two successive months. The reason behind this here is the base theoretical value, which assumes that 1 kWp of PV system produces 146 kWh/month. This results in a very low (PR in winter months where the solar radiation is low, and very high PR (over 1) in the summer semester where solar radiation is high. In this research we wanted to show the real metrics that are used to evaluate these systems so as to perfectly reflect the current situation.

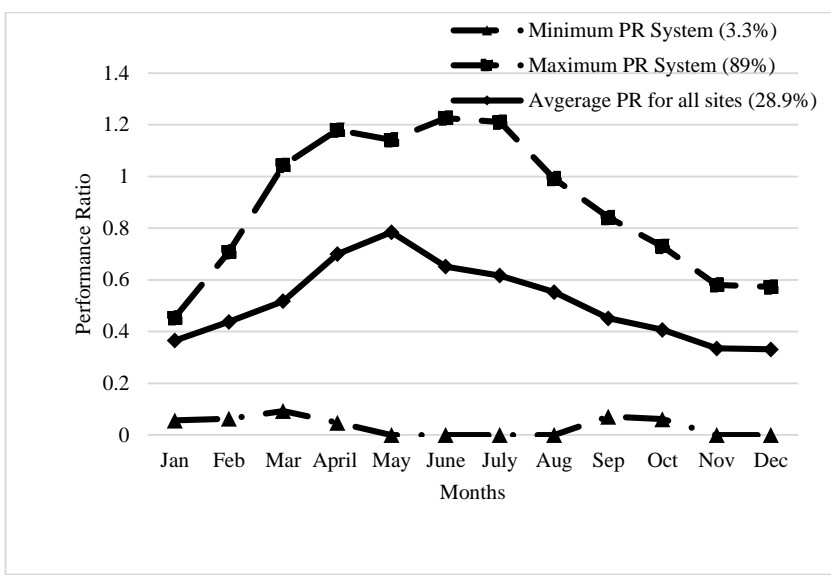

**Figure 6.** Monthly performance ratio for the selected PV systems.

However, this does not mean that this is an accurate way of calculating the PR for the PV systems. The base of PR measure in Figure 6 depends on an assumption that PV systems generate an equal average of energy all over the year. This actually does not reflect the real situation, as a PV system generates different values of energy all over the year depending on the available solar radiation. This explains why the PR in Figure 6 varies very much, whereas the PR in July is the triple of the PR in January. On the other hand, this is not the accurate way of calculating PR, where PR is the ratio between the actual performance as compared to the theoretical or datasheet based performance. PR is usually calculated in terms of system efficiency and for PV systems it is usually in the range of 70–85%. However, the current PV operators in Palestine follow this way of calculating the PR of the PV system as they adapted the method presented by PVsys software which is not accurate. Here, the reason behind posting this figure adapting the wrong practice followed in Palestine is first to reflect the current situation and evaluation metrics that are used in technical and financial studies. Secondly the authors of this research do not have enough data to perfectly calculate the performance ratio of PV systems in Palestine.

From these figures the indicators for some systems are fine, but in some cases there is a dire need for investigation as these system are not operating well. For detailed information Table 4 shows detailed technical and economic data of each site, including system capacity, YF, SPP and yearly income depending on the tariff rate. It is assumed that any system produced more than 1350 kwh/kwp during the year is somehow normal otherwise the system needs investigation. From the table about seven system needs investigation.

**Table 4.** Evaluation of the selected PV systems.

| Site | System Size (kwp) | Annual Production (kwh) | Annual YF (kwh/kwp) | Annual Revenues (ILS) | SPP (Years) |
|------|------|------|------|------|------|
| 1 | 7 | 6144 | 878 | 3809.3 | 11.577 |
| 2 | 10 | 18,157 | 1816 | 11257 | 5.596 |
| 3 | 7 | 11,690 | 1670 | 7248 | 6.085 |
| 4 | 8 | 8007 | 1000 | 4964.3 | 10.152 |
| 5 | 6 | 6791 | 1132 | 4210 | 8.978 |
| 6 | 7 | 8422 | 1203 | 5221.6 | 8.4456 |
| 7 | 7 | 11,654 | 1665 | 7225 | 6.103 |
| 8 | 8 | 12,701 | 1588 | 7875 | 6.4 |
| 9 | 7 | 4798 | 685.4 | 2975 | 14.82 |
| 10 | 7 | 9908 | 1415 | 6143 | 7.1789 |
| 11 | 14 | 21,205 | 1487 | 13,147 | 6.83 |
| 12 | 7 | 10,621 | 1557 | 6585 | 6.52 |
| 13 | 22 | 30,542 | 1368 | 18,936 | 7.43 |
| 14 | 35 | 43,811 | 1262 | 27,163 | 8.05 |
| 15 | 5 | 3996 | 806 | 2478 | 12.61 |

## 6. Energy Policy Status's Review and Discussion

Based on Section 5, it is very clear that not all of the systems are performing well considering the low YF values and the high payback periods. This makes the usability and the investment in such a sector questionable. Thus, in this section we are trying to discuss the reasons behind this issue. There are many reasons for this problem, and these reasons can be classified into two main categories, namely: structural reasons and behavioral reasons.

### 6.1. Behavioral Reasons

Behavioral reasons can be defined as negative or wrong decisions that end-users make in regards a specific technology (PV system in our case). An example of this type of reason is the end-user attitude toward the technology itself. Furthermore, the lack of the correct information about these systems causes negative decision making as well. Nontechnical information on systems feasibility and reliability may greatly encourage the consumers to invest correctly in this field [19]. In [20], the authors have examined the attitude of Palestinian youth toward renewable energy technology and found it really limited with wrong or misunderstood information regarding PV systems. Examples of these myths are listed below:

- The grid connected PV system has a payback period of 3–5 years;
- PV systems are maintenance free systems;
- The grid connected system can work and supply power during the times of electricity outage;
- The batteries in standalone PV system can be charged and discharged simultaneously;
- In grid connected systems, the customer can drain the current he wants while exporting the unneeded currents to the grid (this is against the current division law whereas the current flows are based on the impedance of the electric circuit. Thus, in the case of a home connected directly to the grid, the thevenin impedance of the grid is much smaller than the thevenin impedance of the home. Thus, most of the current will be flowing to the grid.
- PV penetration to the grid can be 100%
- Palestine can be fully independent in terms of electricity using a PV system.

In fact, this is misleading information which is being promoted by politicians, PV companies, normal people, unexperienced staff of DESCOs and outdated researchers of research centers, that has led many end-users, investors, politicians, and even technicians to have made huge mistakes, as it will be discussed later in the structural reasons subsection.

One more issue that may considered as a behavioral reason is the type of the investment. In fact, many of PV systems in Palestine are being installed based on funds and donations from funding agencies. This is in fact, a form of negative behavior of the beneficiaries as they did not pay their share of system's cost. Thus, the failure of the system does not mean a lot for them. For example, while we were investigating in one the selected PV systems, we have visited a school that has a 7 kWp PV system installed on the roof of the school. This system was totally funded by BMZ, Germany. According to the online monitoring system, the PV system had been inoperable for eight months. When we visited the school and asked the principal about the system, he said that the system is very fine and because of it the electricity bill is zero. Although the system is not working, and the electricity bill of the school is not being paid by the school, as the local council is exempting the school from paying the bill.

Thus, to overcome this issue, successful models of such a technology should be promoted. This can be done by adapting proper design and investment practice. According to [21], the students of a sport faculty in Palestine were more aware of solar thermal systems as compared the students of an engineering faculty (mechanical engineering and energy engineering departments were excluded from the questionnaire). This in fact was because at the roof of the sport faculty, there are is a very good solar thermal system that heats the water of the swimming pool inside the faculty. Thus, these students are in touch with a successful model. After all, the promotion of PV systems in a proper technical and financial way can only be done by addressing the structural reasons, as follows.

As a summary, the following action should be taken when considering an installation of PV system in Palestine:

- Training on managing and operating the system for one of the locals who benefits from the system should be done.
- The concept of "free energy" should not be implemented. Thus, in the case of funded systems the beneficiaries should be sharing the consumed energy or the system and hold financial or managerial responsibility of it;
- Professional monitoring should be done by a governmental entity for all installed systems;
- Schools and local community entities should be autonomous in terms of energy consumption and should be exempted from it;
- In the case of private investment, tax exemption should be more than the project cost;
- Stop conventional awareness campaigns and let the people learn by the benefits they earn;
- Anonymous external evaluation for funding projects should be done;
- Achievements should be based on energy production not installed capacity.

### 6.2. Structural Reasons

In general, any wrong, discouraging or negative practice including polices, laws, acts, investments and any type of activities of public and private organizations is a structural reason. For example, the lack of the technical standards or proper training of DESCOs staff on new technologies are reasons for the failure of PV technologies. In addition, there are some other issues such the nature of the power grid, which is another structural reason. Below is a discussion of the most important reasons for the failure of PV system investments in Palestine.

#### 6.2.1. Grid Infrastructure

As mentioned earlier the distribution grid that powers the Palestinian parts is facing many challenges because of the political situation. These challenges include bad voltage stability, low ampacity, critical power stability and low power quality [13]. These problems lead to many grid outages. Grid outages actually are the main reason of having a grid connected PV system down. There are many planned and unplanned outages that occur by distribution companies and the IEC. Figures 7–9 show outages indicators such as the

System Average Interruption Duration Index (SAIDI), System Average Interruption Frequency Index (SAIFI) and the Customer Average Interruption Duration Index (CAIDI) [22]. According to these figures, the failure of any grid connected PV system is very possible after losing the voltage of the main grid. Moreover, the PV system can be operated again only manually by reaching the site. Thus, such systems should be monitored on a daily basis so as to watch the performance of the system and to take action in the case of any abnormal performance.

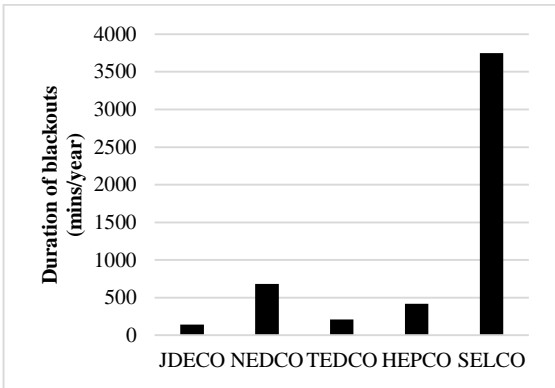

**Figure 7.** Planned and unplanned System Average Interruption Duration Index (SAIDI) in the West Bank 2018.

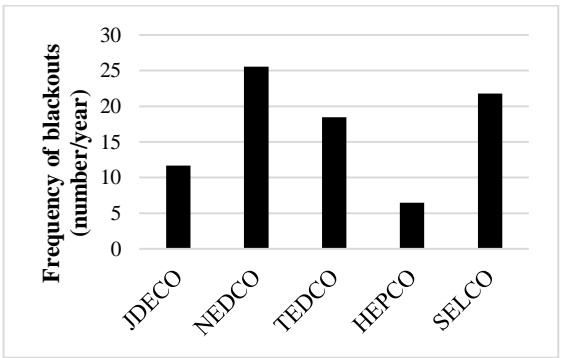

**Figure 8.** Planned and unplanned System Average Interruption Frequency Index (SAIFI) in the West Bank 2018.

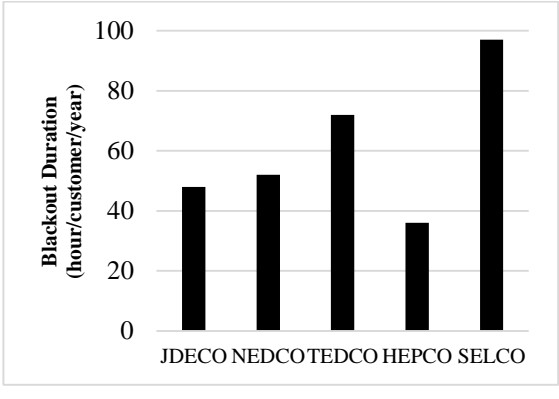

**Figure 9.** Planned and unplanned Customer Average Interruption Duration Index (CAIDI) in the West Bank 2018.

6.2.2. Lack of Training and Technical Standards

In general, there is an absence of technical standard and codes for PV system installation. This leads to a huge deviation in installation practices between operators. In the meanwhile, the controlling behavior of electricity by the Israeli government pushed the Palestinians to think seriously about distributed generation where local generation units based on the PV system can be used at the low voltage network so as to mitigate the electricity shortages. Palestinians consider this technology as an independency tool to abandon the Israeli electricity. It is an actual hope for everyone in Palestine to have their own electricity network [23]. Here, this issue drives the Palestinian to work without serious planning of PV system investments. Politicians become the source of information of this technology. On the other hand, there is a lack of proper training or DESCOs, PENRA and PETL staff on this new technology. This led to either standing against the installation of this technology, as in the case of NEDCO which does not allow any connection of PV system (large systems) to its network, or making wrong decisions such as what happened with TEDCO.

In general, the lack of experience in the field of distributed generation at the Palestinian side, the dire need for electricity and the loose energy polices led to very critical situation fort the Palestinian side of electricity network. Moreover, the exaggerated profits from renewable energy which are based on feasibility studies that were conducted by unexperienced investigators pushed the government, investors and funding bodies to invest in this field which makes the situation more critical. Table 5 shows details about installed, ongoing and proposed PV system capacities in Palestine according to PENRA.

**Table 5.** Details on PV system investments up to 2020.

| PV Project Name | Capacity (MWp) | Governorate |
|---|---|---|
| Operational (39 MWp) | | |
| Ajja Solar Project | 2 | |
| Maithalon | 3 | Jenin |
| Ya'bad Municipality | 4 | |
| Maslmani Co. | 3 | Tubas |
| CocaCola Factories in WB | 2 | - |
| Noor Jericho, PIF | 7.5 | Jericho |
| Hajla Solar Project | 1.5 | |
| Jayous Project | 1 | Qalqilya |
| Palestinian Initiative Program | 5 | - |
| Net Metering Projects | 10 | - |
| Under development (92.8 MW) | | |
| GIE/PADICO | 7.3 | Gaza |
| Noor Tubas, PIF | 9 | Tubas |
| Kafa'a Co. Solar Project | 5 | |
| Noor Jenin, PIF | 3 | Jenin |
| Bani Ne'em | 30.5 | Hebron |
| Askar | 1 | Nablus |
| Beit Forik | 1 | |
| Birzit University | 1 | Ramallah |
| Schools Project | 35 | - |
| Proposed (20 MWp) | | |
| PALTEL | 10 | Jericho |
| Jericho development committee | 10 | |
| Czech project No. 1 | 2 | Tubas |
| Czech project No. 2 | 2 | Tubas |

Based on this table many licenses have been issued to invest in this field without a real policy model that governs this process. For example, by a simple investigation for Tubas

governorate, it can be noticed that the capacity of the licensed PV systems is more than the capacity of the network itself. In the Tubas governorate the grid is managed by TEDCO. The capacity of the distribution grid there is about 20 MVA, meanwhile, the licensed PV based distributed generation from Table 5 is about 21 MWp (2 MWp are under the net meeting project). This causes a huge overlapping authority between PENRA, PETL and TEDCO. They started blaming each other for such a problem following complaints from the investors. Investors want to install systems while this may damage the grid. It is worthy to mention that none of these systems owners were asked to present a grid impact study before getting the licenses.

On the technical side, increasing the level of distributed generation at the Palestinian side of the TEDCO network causes under loading of the Israeli power stations and affects the power quality negatively. Figure 10 shows the current situation of Tubas governorate distribution networks with 10 MVA distributed generation units out of 20 MVA as a maximum capacity of the network.

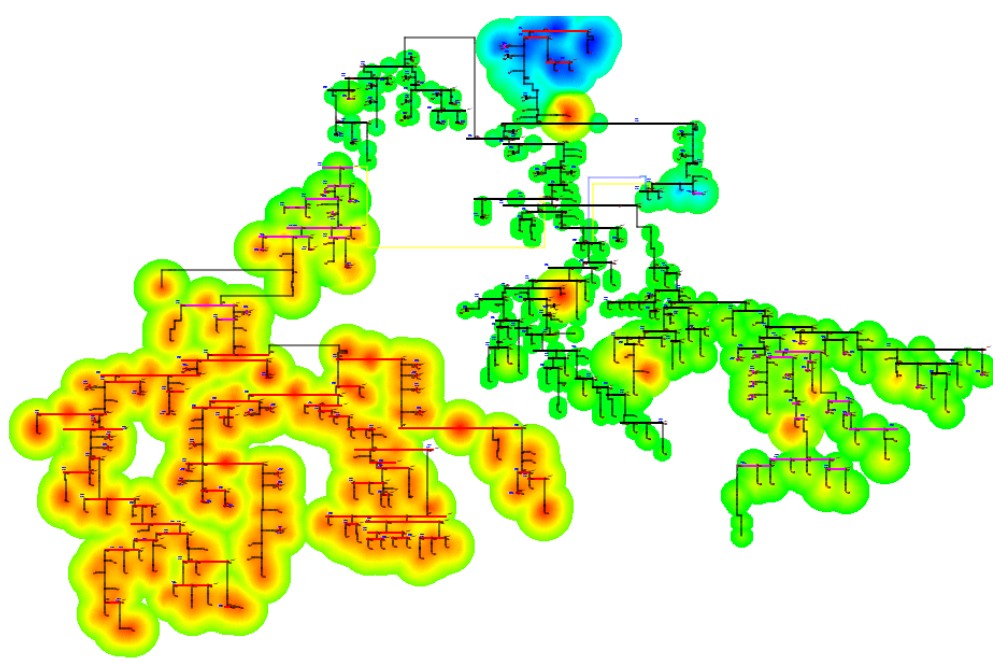

**Figure 10.** Tubas electrical network status with 50% PV generation share.

This figure shows that the network still needs more support (the reddish–yellowish part) as it is overloaded. Meanwhile, the blueish part shows the Israeli connection points which is under loaded with a percentage of 57%. It is worth mentioning that there are still 12 MW licensed distributed generation systems that are waiting to be connected, which will be a real disaster for the Palestinian and Israeli networks.

This is to say that proper training, technical standards and correct technical information should the first step in any investment of PV system in Palestine.

### 6.2.3. Discouraging Policies

In general, there are some discouraging policies by the government for PV system investments, as shown below:

- Net meeting law: according to Decision No, 4/77/17, all the excess energy yield is stored virtually for only 12 months. Afterwards it is deducted from the customer. In other words, starting from March, the customer pays for the excess energy that results from deducting the energy consumption from the renewable energy production. In the case of having excess renewable energy, this amount is transferred to the next month and is valid for 12 months only;

- PV system investment priority is being given to large investors as compared to small corporations;
- DESCOs are not allowed to invest directly in PV system generation;
- There is no unified installation code and standard for PV system installation. Additionally, there is no control for PV system equipment quality in Palestine;
- Licensing procedures are not correct whereas grid impact studies for large systems are being asked after approving the proposal;
- There is an overlap of authority between PENRA, PETL and DESCOs.
- There is no procedure for certifying PV systems providers. Additionally, there is no procedure for enforcing consultancy services with no methodology for certifying consultants for PV systems;
- There is no following up for funded PV system for governmental entities;
- Tax reduction including income tax, value added tax and customs are in some cases more than system costs.

### 6.2.4. Bundled Transmission Lines

In addition to all of the aforementioned reasons for the failure of the PV system in Palestine, the political situation is still considered the main reason for it. As a fact, the lack of a high voltage transmission line is not allowing the Palestinians to form any large investments in PV systems.

For Israel it is not acceptable to not provide electricity to the Palestinians, because Palestinians are about 4 million customers to the IEC. Moreover, the IEC will accept high penetration of renewable energy at the distribution level in the Palestinian territories due to technical and economic reasons. Such a situation will lead to critical power, voltage and frequency stability in the grid. Moreover, IEC will be then providing a free source of voltage for the Palestinians to synchronize their systems without making any benefit. Thus, Israel can unbundle parts of the 161 kV transmission line that crosses the west bank and sell it to the PETL. This will allow the company first to organize the sector of the energy in Palestine, control all of the distributed generation activities and provide a more reliable and stable grid for the Palestinian side.

Israel can make profits as well out of this practice. Historically, electricity was used as a discriminating tool according to the Ronen Shamir book in [24]. The energy was called Israeli energy and boycotted based on that. Now, although the Palestinians were really affected by this decision, the Israelis were affected as well. The Israeli power house at that time faced a lot of troubles in selling their products [24]. After all, this has prompted Pinhas Rutenberg to accommodate his Arab customers so as to sell his product.

Today, Israel can give transmission lines to the Palestinians, while Palestinians can utilize RE energy in generating power. Here, although Israel will lose a share of electricity sales, but this can be compensated by wheeling charges. On the other hand, the IEC will have a much more reliable and stable transmission line with better power flow, power quality and less power losses.

### 7. Conclusions

In this research the current energy policy model for photovoltaic generation in Palestine and the challenges facing it were studied. For this purpose, 15 photovoltaic systems were selected and evaluated based on technical and economic criteria. The typical performance of photovoltaic systems in Palestine was concluded based on this evaluation. According to results the average yield factor of photovoltaic systems in Palestine is in the range of 1368–1816 kWh/kWp per year with a payback period of 5.5–7.4 years. However, the evaluation campaign showed as well that 47% of the selected systems are not working properly and thus classified as failure cases. It was concluded in this research that this large percentage of failure for PV system is due to a number of behavioral and structural reasons. The low awareness and lack of non-technical information were found to be the

main behavioral reasons, while grid infrastructure, lack of technical standards and staff training as well as loose and discouraging polices are the most dominant structural reasons.

**Author Contributions:** Conceptualization, T.K.; methodology, T.K., A.B., H.A. and S.M.; validation, T.K., A.B., H.A. and S.M.; formal analysis, T.K., A.B., H.A. and S.M.; investigation, T.K., A.B., H.A. and S.M.; writing—original draft preparation, T.K., A.B., H.A. and S.M.; writing—review and editing, T.K.; supervision, T.K.; project administration, T.K.; funding acquisition, T.K. All authors have read and agreed to the published version of the manuscript.

**Funding:** This research was funded by An-Najah National University, Grant No. ANNU-1920-Sc005.

**Institutional Review Board Statement:** Not applicable.

**Informed Consent Statement:** Not applicable.

**Data Availability Statement:** Data is available with authors upon request.

**Conflicts of Interest:** The authors declare no conflict of interest.

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
