# Peer review of "Palestine Energy Policy for Photovoltaic Generation: Current Status and What Should Be Next?"

_sustainability, doi:10.3390/su13052996_

Round 1
Reviewer 1 Report
The article discusses the problems related to the functioning of photovoltaic installations in Palestine. The political, legal and technical aspects related to the operation of the energy system are characterized. The structural, legal, political, social and technical barriers affecting energy security of electricity consumers were discussed in detail. Due to the special situation related to the distribution of energy in Palestine, the issues presented in the article are topical and very interesting for the readers.
Detailed comments:
- Please declare all abreviations and symbols in a table of nomenclatures.
- Line 154- Figure 1 and line 185 - Figure 1. The two figures have the same number.
- No reference in the text to Figure 1 from line 154 before Figure 1.
- Errors in the descriptions of figures and tables (lower case, no dots),
- Figures 4-6 - descriptions not readable
In further research:
Electricity storage can be considered. They significantly increase the efficiency of photovoltaic installations. The disadvantage, however, is the significant increase in costs of the installation.
Author Response
Dear Editor and reviewer. Thank you for your time in reviewing this paper. We have considered all reviewer 1 points without any exception.
1) the article discusses the problems related to the functioning of photovoltaic installations in Palestine. The political, legal and technical aspects related to the operation of the energy system are characterized. The structural, legal, political, social and technical barriers affecting energy security of electricity consumers were discussed in detail. Due to the special situation related to the distribution of energy in Palestine, the issues presented in the article are topical and very interesting for the readers.
[ANS]: Thank you for the thumps up.
2) Please declare all abbreviations and symbols in a table of nomenclatures.
[ANS]: Thank you for your comment. We totally agree with your recommendation. However, according to MDPI format abbreviations should be only cited in text after being defined for the first time, not as a table on the beginning of the article. Thus, please allow us to keep it this way. However we have reviewed our article to make sure that all abbreviations have been defined at the first time.
3) Line 154- Figure 1 and line 185 - Figure 1. The two figures have the same number.
[ANS]: Thank you for your comment. We have revised the caption of figure 3
4) No reference in the text to Figure 1 from line 154 before Figure 1.
[ANS]: Thank you for your comment. The citation of the figure is in line 147 just 5 lines before the figure position
5) Errors in the descriptions of figures and tables (lower case, no dots),
[ANS]: Thank you for your comment. The captions of all figures have been checked and revised
6) Figures 4-6 - descriptions not readable
[ANS]: Thank you for your comment. We also agree that the legends of these figures are too small, However, these figures are editable thus, in case that the paper is accepted we will make sure with the production editor that legends and axis labels are readable as the current layout of the paper is not final and making them just bigger now does not help while the best is to control them during production.
7) Electricity storage can be considered. They significantly increase the efficiency of photovoltaic installations. The disadvantage, however, is the significant increase in costs of the installation.
[ANS] Thank you for your comment. That is very true, however, considering the current political situation, the imports of storage units for Palestine is controlled meanwhile, the size of the distribution networks makes the installation of storage units very challenging in addition to the drawbacks you have mentioned. We have add such interesting idea to the revised paper.
Reviewer 2 Report
- The energy crisis referred in [1] had happen? Because as you wrote it, it looks like is gonna happen, but the cited reference is from 2007.
- The share of renewable sources was of 0.9% in 2016, according to [2], while previously you are referring to the consumption from 2018.
- You have like three consecutive paragraphs starting with “on the other hand” (lines 34, 35 and 39). I suggest you to reformulate at least two of them.
- On line 79 perhaps you are referring to the level of income.
- The resolution of Fig.1 and Fig.2 is very poor and the written text is hardly visible. I suggest better quality pictures.
- In line 185 there should be Fig. 3, not Fig. 1 (the one with current situation of energy frame).
- In the paragraphs which starts on line 244, the word “this” should be deleted from the beginning.
- In expression (2) “monthely” should be written “monthly”.
- In the paragraph between lines 269-272 you are stating that the performance factor – PR is independent of the environmental conditions, while in the paragraph after expression (3), between the lines 277-282, you are saying that there are many factors affecting the value of the performance, such as the cell temperature. But the cell temperature is directly affected the ambiental temperature, thus is dependent on the environmental conditions. In this way you are contradicting yourself. So, I think these paragraphs must be properly clarified.
- I did not understand what do you mean by “adapted PV systems”. The entire study case is developed considering “15 PV systems which were adapted” (line 294). Please explain what adapted PV systems means.
- In Figure 4 caption you are refering to yield factor YF, while inside the figure itself is written Capacity factor.
- How come the Performance ratio has values over 1 in Figure 6? As far as I know this parameter is always smaller than 1 (lower than 100%).
- I think that the unit measurement of the CAIDI index represented in Fig.9 is not correct. CAIDI means customer average interruption duration, but from figure 9 results only the number of the annual interruption per customer.
- I don’t understand how the excess of energy yield is stored only for 12 months (line 487-488). Please clarify this aspect.
- In line 510 perhaps 4 M means 4 million customers.
- Last section should be noted as 7 and also is the second section of the paper which is named Conclusions (line 530).
For me is no surprise that the PV systems are not operating as they should, considering the current state of the power grid. It is very difficult to come up with a feasible technical solution while the political factor is the main issue of the entire problem.
Author Response
Dear Editor and reviewer. Thank you for your time in reviewing this paper. We have considered all reviewer 2 points without any exception.
1) the energy crisis referred in [1] had happen? Because as you wrote it, it looks like is gonna happen, but the cited reference is from 2007.
[ANS]: Thank you for your comment. We are so sorry for the misunderstanding, the reference actually for fact that Palestine is suffering from a lack of electricity according to [1]. However the crisis expectation is ours, we have revise the reference location.
2)The share of renewable sources was of 0.9% in 2016, according to [2], while previously you are referring to the consumption from 2018.
[ANS]: Thank you so much for bringing this issue to our attention. There were a reference missing. We have add it and fix the issue.
3) You have like three consecutive paragraphs starting with “on the other hand” (lines 34, 35 and 39). I suggest you to reformulate at least two of them.
[ANS]: Thank you for your comment. We have rephrase the paragraphs
4) On line 79 perhaps you are referring to the level of income.
5) The resolution of Fig.1 and Fig.2 is very poor and the written text is hardly visible. I suggest better quality pictures.
[ANS]: we are sorry for this experince. Anyway, In case the paper is accepted, we will provide scalable images of these figures to production editor whereas the image will be probably posted eventually
6) In line 185 there should be Fig. 3, not Fig. 1 (the one with current situation of energy frame).
[ANS]: Thank you for your comment. This issue is fixed.
7) In the paragraphs which starts on line 244, the word “this” should be deleted from the beginning.
[ANS]: Thank you for your comment. This issue is fixed.
8) In expression (2) “monthely” should be written “monthly”.
[ANS]: Thank you for your comment. This issue is fixed.
9) In the paragraph between lines 269-272 you are stating that the performance factor – PR is independent of the environmental conditions, while in the paragraph after expression (3), between the lines 277-282, you are saying that there are many factors affecting the value of the performance, such as the cell temperature. But the cell temperature is directly affected the ambiental temperature, thus is dependent on the environmental conditions. In this way you are contradicting yourself. So, I think these paragraphs must be properly clarified.
[ANS]: Thank you for bringing this issue to our attention. For sure PR is dependent on environmental issue we have fixed this line which posted by fault.
10) I did not understand what do you mean by “adapted PV systems”. The entire study case is developed considering “15 PV systems which were adapted” (line 294). Please explain what adapted PV systems means.
[ANS]: Thank you for your comment. Nothing specific actually as it means the selected systems in this study. To address your concern we have replace the word adapted to selected in all of the article.
11) In Figure 4 caption you are refering to yield factor YF, while inside the figure itself is written Capacity factor.
[ANS]: Thank you for your comment. This issue is fixed.
12) How come the Performance ratio has values over 1 in Figure 6? As far as I know this parameter is always smaller than 1 (lower than 100%).
[ANS]: Thank you for bringing this issue to our attention. There is a missing paragraph that is added following your comment as below “From figure 6 it is very clear that the PR values have a lot of problems whereas they are fluctuating all over the year. Meanwhile, for a PV system the PR is expected to be almost the same for two successive months. The reason behind this here is the base theoretical value which assumes that 1 kWp of PV system produces 146 kWh/month this results a very low PR in winter months whereas the solar radiation is low and very high PR (over 1), in summer semester whereas solar radiation is high. In this research we wanted to show the real metrics that are used to evaluate these systems so as to perfectly reflect the current situation.”
13) I think that the unit measurement of the CAIDI index represented in Fig.9 is not correct. CAIDI means customer average interruption duration, but from figure 9 results only the number of the annual interruption per customer.
[ANS]: Thank you for your comment. Actually it shows the duration per customer per year in hours so we think it is ok. Please double check.
14) I don’t understand how the excess of energy yield is stored only for 12 months (line 487-488). Please clarify this aspect.
[ANS]: thank you for your comment. We have explained this issue better as below “In other words, starting from March, the customer pays for the excess energy that is results from deducting the energy consumption from the renewable energy production. In case of having excess renewable energy, this amount is transferred to the next month and is valid for 12 months only. “
15) In line 510 perhaps 4 M means 4 million customers.
[ANS]: Thank you for your comment. Yes, we have revised it as you recommended.
16) Last section should be noted as 7 and also is the second section of the paper which is named Conclusions (line 530).
[ANS]: Thank you for your comment. This issue is fixed.
Round 2
Reviewer 2 Report
- I think is incorrect to assume that 1 kWp of PV will produce 146 kWh month. Perhaps this is true for a certain month of a year, but the output of a PV system is influenced by the value of irradiation which is obviously higher in summer and lower during winter. So, from my perspective, is incorrect to determine PR index when considering that the “Calculated nominal plant output” is constant throughout the entire year, because the term from the numerator, the Actual reading of output power, cannot be higher than the calculated nominal output (When a generator produces more than its nominal output?). So, you should rethink the way this PR index is evaluated. I suggest you to determine the PR index on a monthly basis, evaluating the nominal output power depending on the actual values of the monthly irradiation and after that, using the real measured output of the PV systems, you could determine the PR values, and you can also determine the average value for the entire year. In you proceed in this sense than obviously the graphs from Fig.6 would appear different.
- If the unit measurement of the CAIDI index is hours per year, then in Fig.9 you should replace numbers (in fact you wrote “nuber”) with hours, otherwise some may think that customers from JDECO have experienced like 50 interruption per year instead of 50 hours.
Author Response
Dear editor and reviewer we have revised our article based on your new comments without any exception. Please find our response below,
1) I think is incorrect to assume that 1 kWp of PV will produce 146 kWh month. Perhaps this is true for a certain month of a year, but the output of a PV system is influenced by the value of irradiation which is obviously higher in summer and lower during winter. So, from my perspective, is incorrect to determine PR index when considering that the “Calculated nominal plant output” is constant throughout the entire year, because the term from the numerator, the Actual reading of output power, cannot be higher than the calculated nominal output (When a generator produces more than its nominal output?). So, you should rethink the way this PR index is evaluated. I suggest you to determine the PR index on a monthly basis, evaluating the nominal output power depending on the actual values of the monthly irradiation and after that, using the real measured output of the PV systems, you could determine the PR values, and you can also determine the average value for the entire year. In you proceed in this sense than obviously the graphs from Fig.6 would appear different.
[ANS] we totally agree with you actually. This is not a recommended way of calculating PR/ However, we just wanted to follow the methodology that the operators of these system use. This explains actually why is the PR in some months is very low such as in winters months. In general this is not an accurate definition of PR value whereas the actually and recommended PR values should be calculated based on system efficiency which makes PR has the same value all over the year if everything with the system is fine. Thus, our suggestion is elaborate more on this figure and explain that this is the not the recommended way of calculating PR value so as to address your and our concern. Meanwhile, we keep this figure just to show the wrong practice of evaluating these systems which for sure shows misleading results. To address your concern, we have added four paragraphs colored blue so as to show results in better way. Please check the revised paper. We hope this is fine with you.
2) If the unit measurement of the CAIDI index is hours per year, then in Fig.9 you should replace numbers (in fact you wrote “nuber”) with hours, otherwise some may think that customers from JDECO have experienced like 50 interruption per year instead of 50 hours.
[ANS]: Thank you so much for bringing this to our attention. You are totally right. We have revised the figure accordingly.